# The Thermal Stress Coping Network of the Nematode *Caenorhabditis elegans*

**DOI:** 10.3390/ijms232314907

**Published:** 2022-11-28

**Authors:** Eleni Kyriakou, Eirini Taouktsi, Popi Syntichaki

**Affiliations:** 1Laboratory of Molecular Genetics of Aging, Biomedical Research Foundation of the Academy of Athens, Center of Basic Research, 11527 Athens, Greece; 2Department of Biotechnology, Agricultural University of Athens, 11855 Athens, Greece

**Keywords:** heat shock proteins, thermotolerance, *C. elegans*, HSF-1, thermosensation, stress resistance

## Abstract

Response to hyperthermia, highly conserved from bacteria to humans, involves transcriptional upregulation of genes involved in battling the cytotoxicity caused by misfolded and denatured proteins, with the aim of proteostasis restoration. *C. elegans* senses and responds to changes in growth temperature or noxious thermal stress by well-defined signaling pathways. Under adverse conditions, regulation of the heat shock response (HSR) in *C. elegans* is controlled by a single transcription factor, heat-shock factor 1 (HSF-1). HSR and HSF-1 in particular are proven to be central to survival under proteotoxic stress, with additional roles in normal physiological processes. For years, it was a common belief that upregulation of heat shock proteins (HSPs) by HSF-1 was the main and most important step toward thermotolerance. However, an ever-growing number of studies have shown that targets of HSF-1 involved in cytoskeletal and exoskeletal integrity preservation as well as other HSF-1 dependent and independent pathways are equally important. In this review, we follow the thermal stimulus from reception by the nematode nerve endings till the activation of cellular response programs. We analyze the different HSF-1 functions in HSR as well as all the recently discovered mechanisms that add to the knowledge of the heat stress coping network of *C. elegans.*

## 1. Introduction

The roundworm *C. elegans* is a widely used experimental model, gifted with a rapid life cycle, small size, large brood size, self-fertilization mode of reproduction, convenient and cheap laboratory maintenance and storage, transparent body, invariant number of somatic cells, and many other characteristics [1] that are considered advantages for biological studies. Being an ectotherm, *C. elegans* cannot regulate body temperature, which instead depends on ambient temperature [2]. Therefore, the nematode is affected by temperature fluctuations and can detect temperature changes [3]. Thermosensation and thermal stress response in a multicellular organism, such as *C. elegans*, are simultaneously cell-autonomous and cell non-autonomous processes. Coordination of events occurring in differentiated cells and tissues after exposure to high temperatures, is vital for the re-establishment of balance following a stress condition [4,5]. Upon exposure to heat stress, proteins can misfold and form aggregates, which together with cellular mis-organization, organelle dysfunction and altered membrane fluidity occurring in these conditions, can potentially cause cell death. In the worm, a rise in temperature is detected with the aid of specific thermosensory neurons that convert warming into electrical signals transducted through a neuronal circuit to orchestrate a cellular response [4,6]. This review follows the route of these signals from the thermal stimulus up to the cellular changes that are elicited by the conserved heat shock response (HSR) mechanism in *C. elegans*. We summarize the neuronal thermal stress sensing and how the signal spreads through thermosensory neurons to the somatic cells of the worm. We also present all known data about the differential gene expression and the molecular mechanisms that are set in motion to cope with thermal stress; all fine-tuned by heat shock factor 1 (HSF-1), a major transcription factor that is crucial for cell survival as it also contributes to more complex physiological processes [7]. HSF-1 is the single worm homolog of the four mammalian HSFs, which are the main regulators of specific defensive proteins, known as heat shock proteins (HSPs). HSPs are primarily induced in response to heat shock but can also be produced in response to a variety of environmental and cellular stresses, as well as diseases, such as neurodegenerative disorders and cancer [8]. HSPs are known to have cytoprotective effects, acting as molecular chaperones that help to refold or stabilize the unfolded proteins, to dissociate the toxic protein aggregates and to drive the misfolded proteins to degradation, thus contributing to organismal physiology and pathophysiology [9]. Therefore, there is a rising interest in deciphering the molecular mechanisms of HSF1 regulation, the interplay between the downstream signaling pathways and its broad role in proteostasis maintenance and thermotolerance.

## 2. The Thermosensory Circuit

The nematode *C. elegans* can usually be found in nature in decomposing plant material [10] and is able to survive and reproduce in gradient temperatures that range from 15 °C to 25 °C [11]. Interestingly, the nematode detects, adapts and behaves in response to changes of even 0.01 °C on a thermal gradient [6,12], through the action of well-defined thermosensory neurons (See Reviews on temperature sensing by Aoki et al. [13] and Glauser [14]).

### 2.1. Detection of Temperature Changes 

Sensing of temperature changes triggers a robust behavioral response in *C. elegans*. Worms tend to move toward the direction of the minimum temperature deviation from the cultivation temperature (Tc), to avoid conditions in which they cannot reproduce [15,16]. This phenomenon is termed thermotaxis [15]. There are four distinct types of thermotaxis analyzed by Hedgecock and Russell [15], depending on the behavior of the nematode when placed on a thermal gradient. When worms move toward warmer or cooler temperatures, their behavior is called positive or negative thermotaxis, accordingly. Isothermal tracking is worm’s movement perpendicular to the gradient, and if worms do not respond to thermal gradients their behavior is characterized as atactic [15]. In the literature, several different thermotaxis behavioral assays have been described. In most of these protocols, the movement of well-fed worms toward their past Tc, on thermal gradients within the physiological temperature range (15–25 °C) is recorded and analyzed [17]. *C. elegans* can also respond to noxious temperatures (above 26 °C to at least 36 °C), generated on thermal gradients or by thermal pulses, and display a stereotypical avoidance behavior [18,19].

Laser ablation, genetic, behavioral and electrophysiological approaches have established that the AFD (Amphid Finger-like Endings D), are the primary thermosensory neurons (See extended reviews by Goodman and Sengupta [20,21]). AFD is a pair of bipolar sensory neurons terminating in ciliated endings, extending to the worm’s nose [22], and allow worms to detect tiny thermal fluctuations of <0.01 °C from 15–25 °C. Additional sensory neurons required for temperature sensing and thermotaxis are the AWC (Amphid Wing Neurons C) and ASI (Amphid Single Cilium I) neurons [16,23,24,25]. Molecular analysis of thermosensation revealed that the membrane potential of AFD neurons is depolarized, triggered by activation of thermoreceptor currents (ThRCs) through a cyclic guanosine monophosphate (cGMP)-dependent signaling cascade [6] (Figure 1).

In particular, three receptor-type transmembrane guanylyl cyclases (rGCs) are located at the sensoring endings (the GCY-8, GCY-18 and GCY-23 that catalyze cGMP synthesis from GTP [26,27]) and are expressed exclusively in AFD [28,29]. In response to elevated temperatures, rGCS-evoked upregulation of intracellular cGMP levels opens the cyclic nucleotide-gated (CNG) TAX-4/TAX-2 channels [30,31] and permits calcium influx and depolarization [32]. Interestingly, exogenous expression of rGCs in other cells or tissues can also confer thermal responses [33]. The resulting calcium influx contributes to thermosensation in multiple ways. One of them is by regulating the neuronal calcium sensor 1 (NCS-1) expressed in AFD neurons, which probably has a dual function in calcium-dependent balancing of cGMP production [34]. NCS-1 is considered to inhibit GCY-8 and/or enhance cyclic nucleotide phosphodiesterase 2 (PDE-2), which hydrolyzes cGMP and contributes to ThRCs fast adaptation [20,34]. The fast phase of AFD adaptation has a timescale of few minutes, is transcription-independent and ensures high sensitivity of AFD neurons to tiny temperature changes. However, prolonged exposure to warmer temperatures evokes a second, slower (approximately 3–5 h) adaptation phase, which is also determined by calcium influx [35]. This involves the phosphorylation and nuclear localization of CMK-1, the *C. elegans* calcium/calmodulin-dependent protein kinase I (CaMKI), which in turn upregulates the expression of rGCs, increases intracellular cGMP levels and confers long-term adaptation [35]. Additional identified targets of CMK-1 that are required for memorizing the Tc in AFD neurons include the Raf pathway [36], and the cyclic AMP-responsive element-binding protein CRH-1/CREB [37].

In response to noxious temperatures exceeding 26 °C, *C. elegans* shows an acute withdrawal reaction, which is under the control of several sensory neurons, including AFD, AWC and FLP (FLaP-like Dendritic Ending) in the head and PHC (Phasmid Neuron C) in the tail of animals [19]. This thermal avoidance behavior is mediated by a cGMP signaling pathway activating the downstream TAX-4/TAX-2 CGN channel within the AFD neurons, and a heat- and capsaicin-sensitive Transient Receptor Potential Vanilloid (TRPV) pathway, specified by OSM-2 and OSM-9 channels, in FLP and PHC neurons [19,38]. FLP neurons are polymodal nociceptors that detect harsh mechanical as well as noxious heat stimuli (specifically, absolute high temperature rather than thermal changes) and trigger an escape/reversal response [39]. This reversal behavior engages several ion channels, involved in detection, amplification, maintenance and termination of the thermal signal in FLP neurons, and neurotransmission to downstream interneurons (see below). Similar to thermotaxis, previous thermal experience can modify noxious heat avoidance through a mechanism of phosphorylation and nuclear translocation of CMK-1, in FLP neurons [40]. Upon prolonged exposure to heat, progressive nuclear accumulation of CMK-1 reduces neurotransmission and FLP-evoked reversal responses, contributing to sensory adaptation [41].

### 2.2. Downstream Signaling of Thermosensory Neurons

Thermal information sensed and stored in AFD neurons is transmitted to the down-stream AIY (Anterior Interneuron Y) interneurons to regulate thermotactic behaviors in the range between 15 °C and 25 °C in *C. elegans* [16,25]. The excitatory neurotransmission from AFD to AIY interneurons is likely mediated by peptides through the dense-core vesicles (DCVs) pathway [42]. The inhibitory neurotransmission of the AFD/AIY synapse is mediated by glutamate release from ADF through EAT-4 (the homologue of mammalian vesicular glutamate transporter, VGLUT), and subsequent reception of glutamate by AIY via the ionotropic glutamate receptor GLC-3 [43,44] (Figure 1).

In contrast to thermotaxis, where AIY are the major postsynaptic interneurons of AFD, noxious heat avoidance response triggered by infrared irradiation to nose tip of worms depends on AIB interneurons that are connected with AFD via gap junctions and with AWC and FLP neurons via chemical synapses [19]. FLP releases glutamate to AVA (Anterior Ventral Process A), AVD (Anterior Ventral Process D) and AVE (Anterior Ventral Process E) interneurons to promote backward locomotion [45]. In a thermal barrier assay, heat avoidance relies on the NPR-1/FLP-21 receptor in the interneuron RMG [38]. The noxious heat escape response of *C. elegans* proved to exhibit high plasticity, modulated by the thermal context as well as the past thermal experience, food and behavioral state of the animals [40,46]. Furthermore, the avoidance response seems to integrate information from several, parallel neural circuits, with partial functional redundancy, in order to allow a robust behavior strategy under dynamic environmental conditions [23]. Selective optogenetic activation of FLP neurons combined with a forward genetic screen identified both mutations in “general” genes that broadly affect avoidance response and mutations in “FLP pathway-specific” genes, such as the ryanodine receptor gene *unc-68* [45]. Additional mutations that impair reversal behavior in response to the optogenetic activation of FLP involve the genes encoding the neuropeptide receptor FRPR-19 and its ligand FLP-14 [47]. It was shown that FLP-14 is produced in the FLP postsynaptic partners, the AVA/AVD/AVE interneurons, and activates FRPR-19 found in both FLP neurons and its downstream interneurons in a positive feedback loop controlling reversal behavior [47].

The AFD neurons are able to integrate temperature information to a neuronal-endocrine circuit that controls lifespan at warm temperatures. Genetic or laser ablation of AFD or AYI interneurons shortens lifespan mostly at 25 °C [48]. The underlying molecular mechanism involves the activation of CRH-1/CREB by CMK-1/CaMKI in AFD neurons, which in turn upregulates the FMRFamide neuropeptide (FLP-6) that acts on AIY interneurons [49] (Figure 1). FLP-6 targets the AIY interneurons and regulates both insulin pathway in intestinal cells and DAF-9/cytochrome P450-dependent sterol hormone signaling in XXX neurosecretory cells, in order to retain normal lifespan of worms at elevated temperatures [49] (for more details about the effects on lifespan see Kim et al. [3]). Conversely, temperature-induced hormonal signaling from peripheral tissues to AFD neurons can influence thermotactic behavior in worms. For example, *hsf-1* mutants display thermotaxis defects, and the expression of HSF-1 in muscles and intestine rescued these defects, through a nuclear hormone receptor NHR-69 estrogen-signaling pathway [50].

### 2.3. Cell Non-Autonomous Regulation of HSR

In response to acute heat stress, the HSR is activated in cell cultures, unicellular organisms [51,52] or in single cells within *C. elegans* irradiated by a laser microbeam [53], suggesting that it is a cell autonomous response. In a multicellular organism though, the proper regulation and coordination of HSR of individual cells, in diverse somatic tissues, is essential for organismal fitness and survival under thermal stress conditions. Exposure of worms to chronic, mild temperature stress (12–24 h at 28 °C) inhibits egg laying and triggers duration-specific remodeling of the transcriptome with little overlap with the canonical HSR [54]. HSR is induced by acute (even few minutes) exposure to higher temperatures (above 28 °C and up to 37 °C) in almost all cells and tissues of *C. elegans* to promote adaptation and survival. Several studies in *C. elegans* have shown that the nervous system regulates HSF-1 activity in HSR in a cell non-autonomous manner. First, mutations affecting the AFD/AIY neurons reduced heat-induced upregulation of HSP genes by HSF-1 in somatic cells [4]. Further studies established that the neuronal control of HSR is mediated by serotonergic signaling, as animals deficient in serotonin signaling exhibit lower expression levels of cytosolic *hsp-70* (a chaperone gene target of HSF-1) upon heat shock [55,56]. Intriguingly, optogenetic excitation of AFD neurons was sufficient to activate HSF-1 in distant tissues even in the absence of heat, through release of serotonin from serotonergic NSM (Neurosecretory Motor) and/or ADF (Amphid Dual Ciliated Ending F) neurons [56]. Likewise, optogenetic excitation of serotonergic NSM and ADF neurons could activate HSF-1 in remote tissues and protect cells from protein aggregation [56].

Protein misfolding and aggregation can be triggered by unfavorable conditions, such as high temperatures, but is also associated with the pathogenesis of several neurodegenerative diseases, such as Alzheimer’s disease (AD), Parkinson’s disease (PD), Huntington’s disease (HD), etc. [57,58]. A common way to track protein aggregations in nematodes is via expansions of polyglutamine (polyQ), typically associated with HD in humans. Worm models of HD expressing disease-related polyQ tract are susceptible to paralysis and pharmacological induction of HSR could suppress this phenomenon, while loss of HSF-1 activity exaggerates polyQ aggregation [59]. Surprisingly, disruption of the AFD neuronal signaling mitigates polyQ aggregation and toxicity in these worms [55] This might be explained by the fact that animals deficient for AFD/AIY function were still able to induce the expression of HSP chaperones cell-autonomously, in tissues experiencing chronic stress due to accumulation of polyQ proteins. Thus, thermosensory neuronal signaling induces organismal HSR in response to acute heat stress, but suppresses cell-autonomous induction of chaperones under chronic stress caused by misfolded proteins, allowing fine tuning of chaperone levels within the tissues of an organism [55]. Interestingly, trans-cellular chaperone signaling between different tissues can communicate local perturbations of proteostasis independent of neural activity, to coordinate HSP90 expression in adjacent cells and tissues [60].

Cell non-autonomous regulation of HSR therefore involves inter-tissue communication, through inter-cellular or trans-cellular factors, and integrates diverse external and integral cues to produce adaptive behavioral responses. Overexpression of HSF-1 in the nervous system was sufficient not only to mount a robust systemic HSR upon heat stress [61], but also to prolong lifespan under normal conditions of growth [61], and to control fat desaturation that supports adaptation of worms to warmer temperatures [62]. Similar to mammals, serotonin production is the signaling mechanism that modulates various stress-induced behaviors in *C. elegans*, even though chemosensory [63] or other neurons [64] seem to participate in activation of the HSR in distal tissues. One such behavior is the increased feeding of worms after a noxious heat stress, which requires the function of an E3 ubiquitin ligase in serotonergic neurons [65]. Additionally, serotonin release by maternal neurons, following an acute heat shock, activates HSF-1 in the germline and accelerates HSF-1-dependent transcription in germ cells through chromatin alterations. This mechanism protects the germline from stress and promotes the survival and stress resilience of the offspring [66,67]. HSF-1 activity is also coupled to the nutrient-sensing insulin/IGF-1 signaling to regulate normal development of the germline [68]. Although little is known about the regulation of germline HSF-1 by heat stress, it was shown that under heat stress, most germ cells did not induce the canonical HSR, and the activity of HSF-1 is rather repressed, by preferential binding of HSF-1 to helitrons (see Section 4), reducing reproduction [68].

## 3. Functional Domains of HSF-1

In *C. elegans,* the *hsf-1* gene is located in chromosome I and the resulting protein is composed of 671 amino acids (data from https://www.wormbase.org, accessed on 1 September 2022), which form a conserved N-terminal helix-turn-helix DNA-binding domain, an adjacent trimerization domain that consists of hydrophobic heptad repeats (HR-A/B), as well as a C-terminal transactivation domain following another heptad repeat (HR-C) (Figure 2). Spontaneous trimerization of HSF-1 is inhibited by HR-C that folds back and forms intramolecular contacts with HR-A/B [7,69,70]. In humans, a transactivation domain, which has been shown to enhance transcriptional activity, is kept inactive by a regulatory domain located between HR-A/B and HR-C domains [71,72]. The regulatory domain is subject to many post-translational modifications [73]. The exact position of the regulatory domain in *C. elegans* HSF-1 has not yet been identified but is presumably positioned also between the HR-A/B and HR-C domains (Figure 2). *hsf-1* is an essential gene for the nematode and therefore elimination of HSF-1 basic functional domains results in larval arrest, as is the case for the loss-of-function mutant allele *hsf-1*(*ok600)* in which a frameshift deletion results in the absence of the C-terminal HR-C and transactivation domains (380 amino acids eliminated in total) of HSF-1 (Figure 2) [74]. Although the functional domain structure of HSF-1 is highly conserved in eukaryotic species, there are species lacking the C-terminal transactivation domain that regulates the extent of HSF-1 activation and directs HSF-1 to specific target genes. In fact, the only viable *C. elegans hsf-1* mutant allele listed in Wormbase, *hsf-1(sy441)*, is the one missing only the transcriptional activation domain [70,75]. This truncation is caused by a substitution (missense mutation) that leads to a C-terminal elimination of 84 amino acids (Figure 2). The *C. elegans* strain (PS3551) carrying this allele was found to be short-lived, defective in chaperone induction and egg laying, and exhibits a temperature-sensitive larval arrest [7,70]. In view of the foregoing, it comes as no surprise that HSF-1 has functions independent of thermotolerance, including roles in growth, development, metabolism and longevity [74,76,77,78].

In unstressed conditions, mammalian HSF1 exists as a monomer, a state that is preserved through the above-mentioned intermolecular interactions, through post-translational modifications, and through interactions with chaperones [79,80]. Upon heat stress, chaperones are sequestered to be used in protein damage control, thus HSF1 is released, trimerized by coiled-coil interactions between the HR-A/B domains [69] and hyperphosphorylated. Phosphorylation occurs mostly in the regulatory domain, acting like a switch that lifts the restrain off the transactivation domain. In that active form HSF1 is able to bind with high affinity to specific short sequences, termed heat shock elements (HSEs), within promoters of certain target genes and induce their rapid and robust transcription [80]. *C. elegans* HSF-1 has been found in both dimeric and trimeric state upon heat shock [79]. The consensus HSE binding sequence can be generally described as inverted repeats of the pentamer nGAAn, which is true also for the nematode [81,82]. Each HSF-1 moiety of the trimer recognizes one nGAAn repeat and therefore the promoter should host at least three such repeats for efficient HSF-1 trimer binding. The affinity of binding is determined by the number and exact sequence of nGAAn repeats [69] and is found in the majority of *hsp* genes, as well as in a plethora of other genes involved in a broad spectra of physiological cell activities [83]. The diverse HSF-1 functions can be elicited by differences in the composition of factors at HSF-1 binding sites, in the developmental stage of the nematode, in its post-translational modifications and by environmental stimuli [74,80].

Thermal stress has little effect on HSF-1 localization, as in unstressed conditions HSF-1 is already predominantly in the nucleus, yet it significantly affects HSF-1 distribution. Upon heat shock, HSF-1 is allocated to distinct sub-nuclear structures, termed HSF-1 stress granules [84]. These granules are similar to the nuclear stress bodies (nSBs) formed by mammalian HSF1 [85]. Worm HSF-1 stress granules are formed within a minute of heat shock and are dissolved after an hour of recovery. They can be reformed in similar locations if heat stress is reapplied, implying the existence of a scaffold for granule formation [84]. In humans, HSF1 granules are composed mainly of HSF1 bound to heterochromatic pericentromeric satellite repeats (non-coding RNA). Surprisingly, HSF1 bound to HSP promoters was not recovered from human HSF1 stress granules. Thus, their role in HSF1 regulation remains a mystery. *C. elegans* chromosomes do not possess centromeric regions or satellite repeats and at the same time the composition of worm HSF-1 stress granules is largely unknown [84]. A more recent study [86] found that double-stranded RNA (dsRNA) foci partially overlap with HSF-1 stress granules, during heat shock. The dsRNA recovered from these foci contained both sense and antisense transcripts enriched in translation related transcripts and sequences downstream of annotated genes (DoGs). DoG transcripts occur due to reduced efficiency of transcription termination during stress that causes the accumulation of normally untranscribed regions. The authors suggested that dsRNA sequences involved in translation that are over-represented in dsRNA foci, may have a functional role in stress response, i.e., silencing of the respective “translation-related transcripts” to attenuate global translation. Indeed, it has been shown that during heat shock, translation of mRNAs that are not involved in thermal stress response is stalled through pausing of translation elongation and inhibition of translation initiation [51,86,87,88].

## 4. HSF-1 Targets upon Heat Shock

In thermal stress conditions, homotrimerization and binding to HSEs of worm HSF-1 leads to the expression of genes encoding molecular chaperones and other stress responsive genes [7]. At the same time, de novo synthesis of the majority of cell proteins regulating under unstressed conditions is suppressed [51,88]. Interestingly, a recent study showed that the majority of HSEs found in the *C. elegans* genome, reside in transposable elements termed helitrons. The authors showed that helitrons, which comprise ~2% of the *C. elegans* genome, have rendered proximal genes “HSF-1 boundable” or have increased the binding affinity of other genes by supplying more HSEs in the promoter regions [82].

### 4.1. Heat Shock Proteins, the Main Executive Body of HSR

HSPs are molecular chaperones that are upregulated in response to heat shock and attempt to maintain homeostasis in stressed cells by preventing protein aggregation, but they can also function in protein synthesis, processing and degradation [7,80]. HSPs can be divided into two main categories based on their molecular mass: the large ATP-dependent HSPs of 40 to 105 kDa and the small ATP-independent HSPs of 8 to 25 kDa [89,90]. The top 10 genes upregulated by HSF-1 in response to heat shock comprise members of the *hsp-70* family and the small HSPs: *hsp-16.2*, *hsp-16.41*, *hsp-16.11*, *hsp-16.1*, *hsp-16.48* and *hsp-16.49* [7]. The most commonly used gene for reporting HSR is *hsp-70*, although *hsp-16.2* has also been frequently used [78]. The first study of HSR in *C. elegans* [91] identified eight “heat-shock polypeptides”, from which only the 70-kDa polypeptide was ubiquitously expressed. In *Drosophila*, where HSR was first observed, a homologous polypeptide of 70-kDa was also the most prevalent HSP. Despite the early perception that chaperones are the key players of the HSR, more recent studies argue that HSF-1-mediated thermotolerance is not dependent upon the induction of HSPs. This hypothesis was supported by the fact that the hypomorphic mutation *hsf-1(sy441)* did not decrease thermotolerance of the wild-type strain, although it has greatly reduced HSPs mRNA levels [8,92]. However, this HSF-1 mutant strain has given controversial results demonstrating a decrease in heat shock resistance in other studies [4]. The initial thought is that these contradictory results originate from different experimental setups, but it becomes apparent that there is a more complex mechanism underlying HSF-1 function upon heat shock than simply the chaperone activation and course of action [93]. With age, worms lose their ability to produce HSPs, which is probably the reason for the decrease in the effectiveness of HSR in aged nematodes [79,94].

### 4.2. Exoskeleton and Cytoskeleton Integrity Genes

A broad RNA-seq study performed by Brunquell et al. [7] sought the top upregulated and downregulated genes by the *C. elegans* HSF-1. As expected, 9 of the top 15 genes most highly upregulated were HSP genes. Interestingly, this study revealed that among the 654 genes that were significantly upregulated by HSF-1 during heat shock, the functional category of genes with the higher enrichment score were not the *hsp* genes but genes involved in the formation of cuticle structure. In fact, the gene for COL-149, a structural constituent of the cuticle was among the top 15 genes upregulated by HSF-1 in response to heat shock. RNA interference (RNAi) of the expression of another cuticle formation gene, *col-123* was found to decrease *hsp-70* promoter activity, induced by heat shock.

This result was in agreement with a previous study that also highlighted genes involved in the structure and function of the nematode extracellular matrix as main targets of HSF-1 [75]. The calcium-binding protein gene *pat-10* was identified as a direct target of HSF-1 and an essential for thermotolerance gene. In fact, a HSE was found less than 500 bp from the transcription initiation site of *pat-10*. Surprisingly, *pat-10*-dependent resistance to heat stress was not due to upregulation of HSPs. *pat-10* gene is known to be coding a component of the troponin complex that participates in muscle contraction. However, the authors showed that this function of PAT-10 was not responsible for thermotolerance. Furthermore, PAT-10 had been previously shown to be involved in actin cytoskeleton dynamics and endocytosis. Baird et al. [75] showed that heat shock caused muscle filaments to become unorganized and damaged. Thermal stress also resulted in the severe decrease of the ratio of the filamentous actin to globular actin. These consequences were reversed by *pat-10* overexpression. Authors also showed that silencing of *pat-10* by RNAi disrupted endocytosis. The fact that endocytosis is important for thermal stress response was also proven by silencing of the key regulator of coelomocytic endocytosis, *cup-4*, which reduced worm thermotolerance [75]. Remarkably, collagen genes are a major group of genes the expression of which is controlled by HSEs residing in helitrons [82]. *C. elegans* is not the first organism in which cytoskeletal and extracellular matrix genes and in particular collagen genes are upregulated in response to heat stress. Similar observations have been made for fish [95] and even human fibroblasts [96]. In chicken embryos, mice and rats, a heat-shock induced HSP, Hsp47 has been found to specifically bind procollagens in the endoplasmatic reticulum and participate in collagen biogenesis [97]. Hsp47 expression is mediated by HSF1 binding of the HSE in the promoter region of the *Hsp47* gene [97].

## 5. The Post-Translational Fate of HSF-1

The phosphorylation status of mammalian HSF1 as well as other basal or stress-induced post-translational modifications (acetylation, sumoylation, ubiquitinylation, etc.) are associated with the diversification of its functions [73,98,99,100]. Some of these modifications have also been described for *C. elegans* HSF-1 [79,84,101].

### 5.1. Phosphorylation

As a monomer, mammalian HSF1 is constitutively phosphorylated and therefore cannot bind HSEs. In this case, phosphorylation is implemented in order to repress HSF-1 activity [69,102]. In the trimeric state, hyperphosphorylation of serine residues (at least 12) that are mostly located in the regulatory domain, seems to activate the HSF1 complex [73,103]. It has been shown that the number and the location of phosphorylated residues defines the activation state of HSF1. For example, in humans, phosphorylation of S230 and S326 were found to increase the transcriptional capacity of HSF1, while phosphorylation of S303 and S307 were shown to repress HSF1 activation (reviewed by Anckar and Sistonen [73] and Vihervaara and Sistonen [69]). For the worm HSF-1, it has been suggested that phosphorylation may be the most prevalent post-translational modification observed [79].

Under conditions of mild mitochondrial stress HSF-1 is dephosphorylated through the PP2A serine-threonine protein phosphatase complex. In *C. elegans,* PP2A consists of the catalytic subunit LET-92, a single scaffold subunit PPP2R1A/PAA-1 and one of several regulatory subunits that ensure substrate specificity. In this dephosphorylated state, HSF-1 was shown to induce the transcription of small HSPs in a mitochondrial stress-specific manner to protect against protein misfolding, aggregation and toxicity. LET-92 was shown to be the basic component for HSF-1 dephosphorylation but was not required for thermotolerance since hypo-phosphorylated HSF-1 selectively binds HSEs in the promoters of genes needed for mitochondrial perturbation response. This specific combination of genes does not qualify for thermal stress response [104].

### 5.2. Sumoylation

Some post-translational modifications may constitute a requirement for other post-translational modifications to be made. For example, in human HSF1, phosphorylation of S303, mentioned above, serves as a signal for sumoylation at K298. In fact, HSF1 was the first transcription factor in which a phosphorylation-dependent sumoylation motif (PDSM) was discovered. This motif consists of a sumoylation consensus sequence and a phosphorylation consensus sequence joined together. SUMO proteins are known transcriptional suppressors and therefore sumoylated K298 inhibits HSF1 transcriptional activity (reviewed by Anckar and Sistonen [73] and Vihervaara and Sistonen [69]). In *C. elegans*, HSF-1 was also found to be target of sumoylation as it was identified among the 248 proteins conjugated to SUMO [105].

### 5.3. Acetylation

Some HSF1 is also subjected to stress-responsive acetylation of lysines primarily in the DNA-binding domain or domains involved in trimerization and localization. Consequently, acetylation results in the attenuation of the respective HSF1 abilities and therefore in the inhibition of the HSR [106]. Interestingly, members of the sirtuin family of nutrient sensor proteins enhance HSR through deacetylation of the DNA-binding domain of HSF1. This alters the duration of DNA binding thus increasing the transcription of target genes [101]. CCAR-1 was found to be the worm homologue of human CCAR2, which is a negative regulator of the sirtuin member SIRT1 that is a well-established deacetylase of HSF1 in mammals. In *C. elegans*, CCAR-1 was also proven to negatively regulate HSR by decreasing HSF-1 acetylation, mediated by deacetylase activity of the worm homologue of human SIRT1, SIR-2.1, and by increasing HSF-1 binding to *hsp-70* promoter [101].

## 6. Other Factors and Pathways That Aid Coping with Thermal Stress

### 6.1. The Insulin/IGF-1–Like Signaling (IIS)

The insulin/insulin-like growth factor signaling pathway is a highly conserved mechanism involved in development, metabolism, behavior and longevity. In *C. elegans*, insulin-like ligands are bound by the insulin receptor, DAF-2, initiating a downstream signal transduction to kinases AKT-1/-2 through phosphatidylinositol 3-kinase AGE-1. Phosphorylation of DAF-16 prevents it from entering the nucleus. In the absence of insulin-like ligands, DAF-16 translocates to the nucleus to promote longevity gene expression [107,108]. The IIS pathway also contributes to survival after a variety of stress stimuli, including thermal stress. Attenuation of IIS can be obtained by mutations in the core components of the pathway, such as the *daf-2*, which confer both longevity and resistance to heat stress [79,92]. In response to heat stress, the c-Jun N-terminal kinase 1 (JNK)-1 directly interacts and phosphorylates DAF-16, promoting its nuclear translocation [109]. Increased thermotolerance of *daf-2* mutants is therefore DAF-16 dependent but HSF-1 independent. Thus, the elevated stress-induced levels of HSPs upon lowered IIS, are probably DAF-16 driven [92]. Consistent with this, upon heat shock, DAF-16 was shown to enter the nucleus in wild-type worms [110].

### 6.2. HIF-1

The master regulator of oxygen homeostasis, hypoxia-inducible factor 1 (HIF1), is a DNA-binding complex consisting of two subunits: HIF1*α* and HIF1*β*. HIF1*α* is apparently the subunit responsible for coping with hypoxia and it has a homologue in *C. elegans* genome, named HIF-1. Upon normoxia, *C. elegans* oxygenase EGL-9 hydroxylates HIF-1, priming it for binding of E3 ubiquitin-ligase complex component VHL-1 that targets HIF-1 for proteasomal degradation [111]. Therefore, oxygen deprivation, *vhl-1-* or *egl-9-*knockdown are all conditions under which HIF-1 is expressed at constitutively high levels. A handful of studies have revealed a connection between HIF-1 and heat response in *C. elegans*. For example, Trenin et al. [112] showed that HIF-1 upregulation is required for heat acclimation in the nematode, as had been previously proven for mammals. In addition, Carranza et al. [113] showed that the thermotolerance conferred by chlorogenic acid, was HIF-1-dependent. They also found that upregulation of HIF-1, either by heat acclimation or by hypoxia-mimicking conditions, increased worm thermotolerance. The authors attributed worm resistance to thermal stress, at least in this case, to the induction of autophagy, mediated by HIF-1 and HSF-1. Autophagy presumably enhances the ability of worms to handle damage produced by thermal stress.

### 6.3. Autophagy

Autophagy (macroautophagy) is an efficient mechanism for the degradation of proteins and cytosolic components in general, thus contributing to cellular homeostasis. Autophagy is initiated in response to various stresses, such as nutrient deprivation, hypoxia, pathogens etc., by the formation of autophagosomes that encapsulate damaged cytoplasmic material. Autophagosomes are then fused with lysosomes and their content is degraded by hydrolases and recycled [114,115]. Recently, thermal stress was added to the list of autophagy triggers. Specifically, it was shown that hormetic heat shock systemically induces autophagy in *C. elegans*, in an HSF-1-dependent manner [114,115]. Hormetic heat stress is defined as a short-term exposure of worms to temperatures that would long-term be lethal, and which brings about induced thermotolerance, i.e., survival under lethal heat shock [116]. HSF-1 seems to play an important role in autophagy induction during thermal stress. In fact, the promoters of many autophagy-related genes contain putative HSEs; but it remains to be clarified whether HSF-1 directly impacts their expression or there is an intervening mechanism. Autophagy genes are required for the increased thermotolerance of worms after hormetic heat shock [117]. Moreover, overexpression of HSF-1 increased the formation of autophagosomes in muscle, neurons and intestines of the nematode [8]. Knockdown of mitophagy (a specific cellular process for elimination of dysfunctional mitochondria by autophagosome engulfment for subsequent lysosomal degradation) genes also reduced heat stress resistance in worms, due to increased protein aggregate formation caused by autophagy downregulation [118]. Autophagy seems to be a more competent method of damage control than HSR in the case of larger protein aggregates and dysfunctional organelles resulting after heat shock [8].

### 6.4. Non-Coding RNAs

MicroRNAs (miRNAs), first discovered in *C. elegans* [119,120,121], are small, non-coding RNAs that control gene expression and a wide variety of physiological process through post-transcriptional silencing. Therefore, it is of no surprise that a process with high complexity such as the HSR would implicate post-transcriptional regulation through miRNAs. Specifically, the *mir-71*, *mir-239*, *mir-80*, *mir-229* and *mir-64-66* cluster, which control survival during heat stress, were identified by high-throughput small RNA sequencing and their function was tested by appropriate deletion strains. For example, *mir-71* and *mir-239* were shown to be required for generating embryos at high temperatures [119]. However, more research is needed on whether these miRNAs are direct targets of HSF-1 and on what are their direct mRNA targets. It is also unclear if the changes in miRNA expression upon heat shock are an intentional result of the thermal stress response or the byproduct of defective transcriptional/post-transcriptional silencing mechanisms upon elevated temperatures. In a more recent study [122], 6 miRNAs were identified to be upregulated by HSF-1 during heat shock: miR-784, miR-231, miR-86, miR-53, miR-47 and miR-34. While 5 of these miRNAs are of unknown function, miR-34 is a highly conserved miRNA that is implicated in ageing-associated processes. Interestingly, miRNAs are not the only type of non-coding RNAs (ncRNAs) involved in heat coping mechanisms. A study by Schreiner et al. [123] uncovered Piwi RNAs, long intergenic ncRNAs, pseudogene- and repeat-derived RNAs and unclassified ncRNAs expressed in *C. elegans* additionally to miRNAs, under heat stress conditions. The authors established a direct link between the expression of ncRNAs, such as miR-239, Helitron1_CE transposons and the pseudogene *dct-10,* and HSF-1. HSEs in the promoters of these ncRNAs are bound by HSF-1 in response to heat shock.

Much ink has been spilled about the transcriptional activation of HSR and the processes involved in safeguarding the proteome. However, less is known about the restoration of HSP gene expression to basal levels after the return to normal conditions. A recent study [124] demonstrated the involvement of the miRNA pathway in *hsp-70* downregulation following heat shock. Upon heat shock, *hsp-70* mRNA is up-regulated over 100-fold and then once normal temperature is restored, these levels drop to baseline within 24 h. In *C. elegans* this rapid restoration of *hsp-70* expression levels was found to depend on the miRNA miR-85 and the Argonaut (AGO) protein ALG-1. According to the authors, the 3′UTR of *hsp-70* harbors two miR-85 binding sites. In unstressed conditions, miR-85 guides ALG-1 to target *hsp-70* mRNA for destabilization through imperfect base pairing. This way *hsp-70* expression levels are diminished to basal. This gene expression reset upon recovery proved to be essential for the survival of heat stressed worms. Apparently, maintenance of extremely high levels of HSPs for a prolonged period of time turns out to be toxic for the organisms.

### 6.5. m^5^C Methylation

Methylation of carbon-5 of cytosines in tRNA, rRNA, mRNA and ncRNA molecules is an evolutionary conserved mechanism performed by m^5^C RNA-methyltransferases, with yet largely unknown functions. Some possible functions are protection from degradation and modulation of translational fidelity [125]. It is also evident that although these chemical marks are often not essential under normal conditions and organisms present subtle phenotypes in their absence, they are involved in responses to environmental challenges. In fact, with the use of a *C. elegans* strain devoid of m^5^C in RNA, a role for m^5^C methylation in thermotolerance was uncovered as mutant worms were temperature-sensitive [126]. m^5^C methylation effect on the nature of HSR was shown to be subtle, with gene expression differences found mainly in genes involved in cuticle and ribosome formation. A more detailed codon-occupancy investigation showed that m^5^C methylation enhanced translation of leucine UUG codon upon heat shock. Interestingly, tRNA Leu-CAA is the only cytoplasmic tRNA with an m^5^C-modified wobble position. Therefore, m^5^C tRNA wobble methylation is probably involved in thermotolerance [126].

### 6.6. Mitochondrial Perturbation

Ageing in *C. elegans* is accompanied by proteostasis malfunction, which occurs partly due to attenuation of the HSR [89]. Labbadia et al. [127] identified mitochondrial electron transport chain (ETC) as an important factor in HSR repression that comes with age. These authors also found that mild ETC perturbation either by genetic knockdown or with the use of xenobiotics, sustained HSR levels in ageing worms. However, not all mitochondrial stresses are able to confer HSR maintenance with age. It was shown that knockdown of subunits of mitochondrial respiratory chain complexes I, III, IV and VI and of mitochondrial protein transport gene *tomm-22* and mitochondrial ribosomal protein gene *mrps-5* significantly increased heat shock resistance, while knockdown of other genes encoding proteins of mitochondrial function, *dnj-21*, *tin-44*, *mrpl-1*, *mrpl-2*, and *spg-7*, had little or no effect on stress resistance. This mild mitochondrial perturbation was found to promote thermotolerance in an HSF-1 dependent way [127].

Mitochondrial perturbation, accompanied by heat stress resistance was also observed in *lonp-1* deficient *C. elegans* mutants [128]. LONP-1 is the worm homolog of human LonP1, a highly conserved mitochondrial protease that functions as a peptidase, a chaperone and a DNA-binding protein, thereby fulfilling its role as a central regulator of mitochondrial activity [129]. Although LonP1 protease is considered as a stress-responsive gene, it is not temperature-inducible [130]. In our recent study, we created a *lonp-1* knockout *C. elegans* strain and showed that despite the disturbed mitochondrial function and impaired growth, fertility and lifespan in the mutant animals, there was a robust induction of both mitochondrial unfolded protein response (UPR^mt^) and cytosolic HSR. Therefore, the *lonp-1* mutant strain was significantly more thermotolerant compared to wild-type, even with the use of slightly aged (day 3 of adulthood) individuals [128]. However, the responsible factors and mechanisms, or the interplay between them, remains unclear.

## 7. Concluding Remarks

Undoubtedly, the struggle to remain alive and unharmed upon elevated temperatures is a highly complicated process, even in simple invertebrate organisms, such as the nematode. HSF1 is admittedly the most important transcription factor driving the HSR in a plethora of species ranging from yeast to mammals. However, HSR, which is considered as the function of HSPs in response to thermal stress, is not the only necessary event for maintaining homeostasis under the circumstances, as was previously believed. Preservation of exoskeletal and cytoskeletal integrity, also HSF1 driven, is gaining popularity as an equally important process. Moreover, other physiological processes that participate in thermotolerance are being uncovered over time, sometimes dependent and sometimes independent of HSF1. In addition, the variety of components involved that range from transcription factors, chaperones, enzymes, mRNAs, miRNAs, dsRNAs, tRNAs, transposons, ribosomes, autophagosomes to methyl-, acetyl- and phospho- groups and SUMO-proteins, unveils the complexity of this fine-tuned survival program. The precise interconnections and interplay of all of these components are not yet well-established and therefore future studies should be oriented toward this end. Although arguably, this is the most difficult task, it has become increasingly evident in the recent literature that in a multicellular organism each process is connected to numerous other processes and each molecular or cellular component is simultaneously or successively involved in numerous processes and is able to compensate for others when a mishap occurs. Thus, all findings should be skeptically examined under this prism. Modern high throughput techniques as well as the use of appropriate, suitable and convenient animal models, such as the *C. elegans*, will certainly help with deciphering the complicated processes of organismal stress responses.

## Figures and Tables

**Figure 1 ijms-23-14907-f001:**
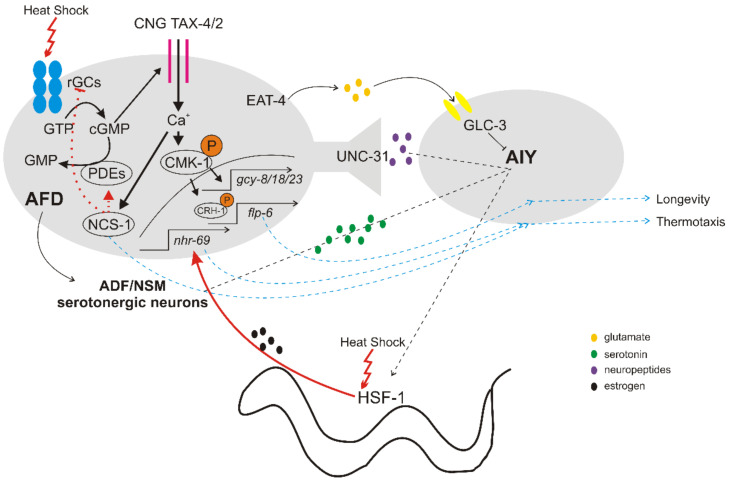
Schematic representation of AFD/AIY thermosensory circuit signaling. Temperature rising is received in AFD sensory neurons by rGCs that upregulate cGMP concentration. This leads to the opening of the CNG TAX-4/TAX-2 channels and the release of a calcium influx, resulting in depolarization of AFD membrane potential. The calcium influx maintains cGMP concentration levels through NCS-1 and further upregulates rGCs via CMK-1 phosphorylation. Phosphorylated CMK-1 enters the nucleus and among regulation of rGCs expression levels, also participates in CRH-1 phosphorylation and subsequent FLP-6 activation that is responsible for normal longevity under elevated temperatures. AFD activation leads to both excitatory and inhibitory signal transmission on AIY interneuron possibly by UNC-31-mediated peptides release and by glutamate release, accordingly. This activation can also release serotonin from serotonergic neurons and upregulate HSF-1 through AIY in the intestine. These signals are necessary for AIY to promote thermal behaviors (thermotaxis) and activate HSF-1-related organismal thermal responses in somatic tissues. In succession, HSF-1 can also signal to AFD in part through estrogen signaling to ensure normal thermotaxis.

**Figure 2 ijms-23-14907-f002:**
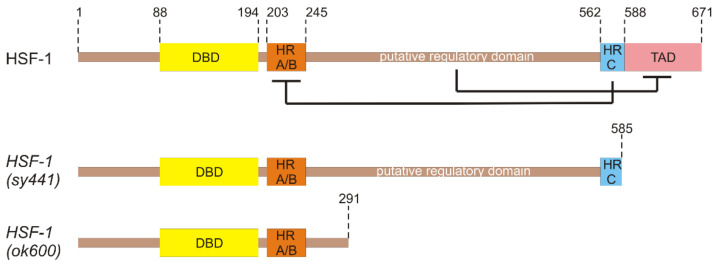
Domain structure of *C. elegans* HSF-1 full-length protein and the truncated alleles *sy441* and *ok600*. Numbers indicate amino acid positions. DBD is the DNA-binding domain, HR-A/B and HR-C are the trimerization domains and TAD is the transactivation domain. HSF-1 trimerization is negatively regulated by intramolecular interactions between the HR-A/B and the HR-C domains. The exact position of the regulatory domain in the *C. elegans* HSF-1 is not yet determined. However, it is most possibly located between the two trimerization domains and negatively regulates the trans-activating capacity of HSF-1. *sy441* allele is lacking the transactivation domain, while *ok600* allele is lacking both TAD and the putative regulatory domain.

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
