# Peer review of "The Thermal Stress Coping Network of the Nematode Caenorhabditis elegans"

_ijms, 2022, doi:10.3390/ijms232314907_

Round 1

Reviewer 1 Report

This review covers the ways in which C. elegans can respond to thermal stress- centering on the role of HSF-1 and its associated partners in this process.  This review does a good job covering the breadth of the information known about HSF-1 and what is known in worms about thermal stress.  However, for this to be a good resource for the community there is editing/re-writing that needs to occur in order to clearly explain the current literature and what is specifically known about these processes in C. elegans.

Major Concerns:

The authors need to spend some time defining the different types of thermal stress and how they are specifically used in C. elegans.  They have a nice definition of hermetic heat shock- but not the normal heat shock protocols that are used to elicit the HSR.  Maybe a table or graph to illustrate the different types of thermal stress commonly used- including the difference between normal thermotaxis and noxious heat avoidance (see comment below).  This information will be especially important of researchers who work on these same systems outside worms or worm researchers in other fields. 

The authors spend a lot of time linking the temperature sensing circuits that have been defined for thermotaxis to the activation of the HSR.  While there are definitely links between these two processes- the authors do not present a clear picture of how these experiments were done and what is really known about the link between thermotaxis and HSR.  Many/most thermotaxis experiments are done in temperature ranges which do not exceed 25-27C, which is not high enough to elicit a canonical HSR.  What may be more relevant in many cases is the circuitry associated with noxious heat avoidance.  

In general, the authors need to go through and do a thorough re-check of their citations.  As this is a review article this is a place people can use a starting point for understanding this field- so there is even higher bar to make sure that anything cited is accurate to the what is being said.  Some examples where different citations are needed: Line 344- that cites Gomez-Pastor et al 2017 (75) which does not at all cover the information in the preceding sentence or Line 283- that cites Labbadia and Morimoto 2015 (70) which is not accurately interpreted in the preceding sentence.  Additionally, there are a number of places where the authors explain a very specific finding about HSF-1 and then cite a review paper. For example, Line 350 the authors discuss activation of Trimeric HSF-1 by phosphorylation and cite Anckar and Sistonen 2022- this is a long and very general review of HSF-1 function.  While Anckar and Sistonen do discuss phosphorylation of HSF-1- they didn’t do any of the research and this makes it a huge lift for the reader to then go back and read additional papers cited by this other review if they want to know about this topic (and to further figure out if this holds true for C. elegans HSF-1, see comment below).

Throughout the document the authors routinely use the format of writing heat-shock factor 1 as HSF-1 and the title of the document and abstract stress that this article is about Thermal Stress in C. elegans.  However, many times in the article the information discussed/cited is from other systems such as mammals or yeast- and has never been shown in C. elegans.  By always writing the protein in the form specific to C. elegans and often not specifying in what organism things are shown in- the authors are suggesting to the reader that things are known about the function of HSF-1 in C. elegans which are not known.  The authors need to either write the protein with the mammalian/yeast designation (Hsf1/HSF1) or clearly specify when information has been specifically shown in another system which is not C. elegans.  An example where this is done much more clearly is the section of Acetylation of HSF-1 and this should be used as a model for the rest of the paper.

In the first paragraph of part 2 (HSF-1 targets upon heat shock).  The authors state that “binding of HSF-1 leads to suppression of de novo synthesis of the majority of cell proteins” and then cite the Labbadia and Morimoto (2015) review.  First, this review does not specifically talk about C. elegans HSF-1 and talks only more generally about HSF1 in many organisms but most specifically mammals and yeast.  Second, there is no suggestion in this paper (or others) that the binding of HSF1 to its targets leads directly to the general decrease in de novo protein synthesis seen heat shock, which is done through RNA splicing inhibition.  In fact, HSF-1 nuclear stress granules may be a site within the nucleus that allows for the specific slicing of RNAs needed for the stress response.  It is true that a general decrease in protein production is seen as part of the HSR- but to link it directly to HSF-1 target binding is either incorrect or a number of steps removed.

In the first paragraph on the section on the regulation of exo/cytoskeleton components the authors interpretation of the data from Brunquell et al. while technically accurate is somewhat misleading.  While the GO term most enriched for HS activated HSF-1 targets was “Cuticle structure”, “response to stress” was also an enriched GO term and 9 of the top 15 genes most highly upregulated by HSF-1 under heat shock were hsp genes.  Thus, while it is interesting that HSF-1 is likely directly regulating genes involved in structural elements- the way this is written it suggests it may be the main role. 

Minor

While this review is focused on somatic responses to heat stress (as the authors point out in the first paragraph)- it would be good to acknowledge that A) there are heat shock responses happening in the germline, B) in adult worms the expression of HSF-1 is very high in the germline, and C) much less is known about how the germline responds to heat shock in comparison to the soma.

The authors should always refer to granules containing HSF-1 as either HSF-1 stress granules or nuclear stress granules.  Generally, the term “stress granule” is reserved for an RNA/Protein granule formed in the cytoplasm, which does not include HSF-1.

Line 64: the survival of C. elegans across temperatures should have a citation.

Line 72: Citation of Reference 6 doesn’t fit with the sentence here- the paper cited deals with the excitation/signaling of neurons with temperature- not thermotaxis which the sentence is about

Lines 202-204: Should be written as follows: “In humans, a transactivation domain, which was shown to enhance 202 transcriptional activity, is kept inactive by a regulatory domain located between HR-A/B 203 and HR-C domains [56, 57].”

Lines 217-219: Should be written as follows: The C. elegans strain (PS3551) carrying this allele was found to be short-lived, defective in chaperone induction and egg laying, and exhibits a temperature-sensitive larval arrest [7, 60].

Lines 242-244: In that active form HSF-1 is able to bind to high affinity specific short sequences, termed heat shock elements (HSEs),  within promoters of certain target genes and induce their rapid and robust transcription [64].

Lines 266-268: Should be written as follows: A more recent study [69] found that double-stranded RNA (dsRNA) foci partially overlap with HSF-1 stress granules during heat shock.

Line 272: The sentence should start with “The authors suggested…”

Lines 316-317: “Interestingly, this study revealed that among the 654 genes that were significantly upregulated by HSF-1 during heat shock, “  There were other genes that were found to be regulated by HSF-1 irrespective of heat shock.

Reviewer 2 Report

Kyriakou et al review IJMS

  This is a fairly comprehensive and well-referenced review regarding the thermal stress response in C. elegans.  I have two general comments and a few minor ones:

General:

  1). The review goes into significant detail regarding the non-autonomous regulation of the C. elegans heat shock response.  Although it mentions in passing that there is a cell-autonomous component to this, little attention is given to it.  Minimally, the authors should mention that a heat shock response (or at least the induction of genes controlled by a heat shock promoter) can be induced in single cells by laser irradiation, indicating that neuronal control of the heat shock response is not required.

2)  The review makes the interesting point that the HSF-1-dependent gene expression response in C. elegans is not limited to chaperone proteins, and in fact cuticle components are a major component of this response.  I think it would be helpful to readers for the authors to comment on whether they believe this is a weird nematode-specific phenomenon (e.g., like operons), or whether there is evidence that the non-HSP response is seen in other organisms.

Minor:

Line 397:  I believe both AKT-1 and AKT-2 phosphorylate DAF-16, and both have to be depleted to induce DAF-16 nuclear localization.

Line 407:  Heat shock-dependent nuclear localization of DAF-16 depends on the JNK pathway, not IIS, so this should be mentioned here.

Line 472-475:  Rephrase this awkward sentence?

Line 494:  "Some indicative ones are protection …". Please rephrase

Line 510: These authors …

Line 537:  "is not the only and presumably not the only .."  Duplicate phrase?

Round 2

Reviewer 1 Report

In the revised manuscript Kyriakou et al highly edited their review article "The thermal stress coping network of the nematode Caenorhabditis elegans."  The changes made greatly improved the clarity and accuracy of the information presented.  I especially appreciated the care taken to clearly describe differences in temperature responses in different temperature spectrums.  Beyond a basic re-check for grammar/spelling and the correct use of HSF-1 vs HSF1 is all that seems necessary for publication.